# Protein Kinase A Controls the Melanization of *Candida auris* through the Alteration of Cell Wall Components

**DOI:** 10.3390/antiox12091702

**Published:** 2023-08-31

**Authors:** Ji-Seok Kim, Yong-Sun Bahn

**Affiliations:** Department of Biotechnology, College of Life Science and Biotechnology, Yonsei University, Seoul 03722, Republic of Korea; wltjrs123@yonsei.ac.kr

**Keywords:** Ras/cAMP/PKA signaling pathway, PKA catalytic subunits, Tpk1, Tpk2, melanization, pathogenic fungi, *Candida auris*

## Abstract

*Candida auris*, a multidrug-resistant fungal pathogen, significantly threatens global public health. Recent studies have identified melanin production, a key virulence factor in many pathogenic fungi that protects against external threats like reactive oxygen species, in *C. auris*. However, the melanin regulation mechanism remains elusive. This study explores the role of the Ras/cAMP/PKA signaling pathway in *C. auris* melanization. It reveals that the catalytic subunits Tpk1 and Tpk2 of protein kinase A (PKA) are essential, whereas Ras1, Gpr1, Gpa2, and Cyr1 are not. Under melanin-promoting conditions, the *tpk1*Δ *tpk2*Δ strain formed melanin granules in the supernatant akin to the wild-type strain but failed to adhere them properly to the cell wall. This discrepancy is likely due to a decreased expression of chitin-synthesis-related genes. Our findings also show that Tpk1 primarily drives melanization, with Tpk2 having a lesser impact. To corroborate this, we found that *C. auris* must deploy Tpk1-dependent melanin deposition as a defensive mechanism against antioxidant exposure. Moreover, we confirmed that deletion mutants of multicopper oxidase and ferroxidase genes, previously assumed to influence *C. auris* melanization, do not directly contribute to the process. Overall, this study sheds light on the role of PKA in *C. auris* melanization and enhances our understanding of the pathogenicity mechanisms of this emerging fungal pathogen.

## 1. Introduction

Infections caused by pathogenic fungi are increasing globally, resulting in a concurrent rise in related fatalities. *Candida auris*, an emerging fungal pathogen first identified in 2009, has rapidly become a significant public health concern worldwide due to its multidrug-resistant properties [1]. It has instigated outbreaks in healthcare facilities across various countries, including the United States, the United Kingdom, India, and South Africa [2]. *Candida auris* can cause a range of infections, including bloodstream infections, wound infections, and otitis externa [3]. Notably, it exhibits significant resistance to existing antifungal agents, resulting in a substantial mortality rate of 30% to 60% among immunocompromised individuals [4]. Compounding the issue, clinical manifestations of *C. auris* infections do not differ markedly from those caused by other *Candida* species, complicating early diagnosis [5]. Given these challenges, a comprehensive understanding of the molecular mechanisms underlying the pathogenicity of *C. auris* has become of paramount importance.

Melanin, an insoluble pigment molecule, is ubiquitous across all living organisms, including animals, fungi, and bacteria [6]. Its primary role is in cellular protection against oxidative stress, UV (Ultraviolet) radiation, and acidic conditions, making it a critical factor for cell survival and environmental adaptation [7]. Various forms of melanin, such as DHN (1,8-Dihydroxynaphthalene) melanin, DOPA (L-3,4-dihydroxyphenylalanine) melanin, and pyomelanin, can be found in pathogenic fungi [8,9]. Notably, *Candida* species like *C. albicans* and *C. glabrata* primarily produce DOPA melanin through the oxidation of catecholamines [10,11]. DOPA melanin production is known to be catalyzed by laccase, and in the context of *C. albicans*, a representative *Candida* species, laccase activity has been detected, albeit minimal. However, a specific gene associated with laccase activity has yet to be identified.

Recent research has uncovered that *C. auris*, an emerging multidrug-resistant pathogen, is capable of producing melanin, akin to other *Candida* species [12]. Interestingly, the melanin granules in *C. auris* do not serve as a virulence factor, as seen in *C. albicans* or *C. glabrata* [12]. Instead, they can be produced using a variety of substrates, such as DOPA, dopamine, epinephrine, and a brain mixture, with the choice of substrate varying depending on the clade [12]. Notably, *C. auris* is unable to use L-tyrosine as a precursor for melanin synthesis and exhibits no laccase activity [12]. Factors such as temperature, pH, and cell density influence the formation of melanin granules, and the cell wall polysaccharide content is key to the successful attachment of these granules to the cell wall [12].

In this study, we aimed to uncover the role of the Ras/cAMP/PKA signaling pathway in the melanization process in *C. auris*. While the core components of the Ras/cAMP/PKA signaling pathway, including the G-protein-coupled receptor (GPCR), RAS-GTPase, adenylyl cyclase, phosphodiesterase, and PKA regulatory subunit, did not significantly affect melanization in *C. auris*, we found that the PKA catalytic subunits, Tpk1 and Tpk2, played a crucial role. Additionally, we demonstrated that these proteins influenced the expression of multiple chitin-synthesis-related genes under melanin-inducing conditions and played a role in the attachment of formed melanin granules to the cell wall polysaccharide content. In summary, our research presents a comprehensive understanding that *C. auris* melanization is regulated by PKA catalytic subunits in a Cyr1-independent manner.

## 2. Materials and Methods

### 2.1. Candida auris Strains and Growth Media

This study employed *Candida auris* strains, including the wild-type strain (B8441/AR0387) provided by the CDC, delineated in Appendix A. We preserved all isolates, including the constructed mutant strains, as frozen stocks at −80 °C in a YPD (Yeast Extract-Peptone-Dextrose) medium containing 20% glycerol. We cultured yeast strains at 30 °C with consistent agitation at 200 rpm in YPD broth (1% yeast extract, 2% peptone, and 2% D-glucose), or on YPD plates that incorporated 2% agar in YPD broth.

### 2.2. Total RNA Preparation and Quantitative RT-PCR

The wild-type and PKA mutant strains of *C. auris* were cultured overnight in 30 mL of YPD broth at 30 °C in a shaking incubator. Following this, the cells were subcultured in 40 mL of fresh YPD broth until they achieved an optical density at 600 nm (OD_600_) between 0.6 and 0.8. The cells were then collected via centrifugation, swiftly frozen in liquid nitrogen, and subjected to lyophilization. For the melanin-inducing conditions, we collected a 10 mL sample as the basal measurement, while we further incubated the remaining 30 mL with the melanin-inducing medium. We extracted total RNA using the Trizol method and used Easy-blue (Intron: Seoul, Republic of Korea) for isolation. This purified total RNA was utilized for reverse transcription to generate complementary DNA (cDNA) with the assistance of reverse transcriptase (Thermo Scientific: Waltham, MA, USA). The resultant cDNA was employed for quantitative PCR with gene-specific primer pairs. We conducted the real-time PCR analysis using the CFX96TM Real-Time system (Bio-Rad: Hercules, CA, USA). We utilized the expression of *ACT1* as an internal control to normalize the results. One-way ANOVA was used for statistical analysis, and Bonferroni’s multiple comparison test was employed to assess the significance between different samples. All experiments were performed in triplicate and included three independent biological replicates.

### 2.3. C. auris Melanization in Liquid Media

To conduct the melanization assay for *C. auris*, we inoculated wild-type and all mutant strains in 2 mL of liquid YPD. These were then cultured overnight at 30 °C in a shaking incubator. Following the cultivation period, all strains underwent two washes with minimal media (15.0 mM glucose, 10.0 mM MgSO_4_, 29.4 mM KH_2_PO_4_, 13.0 mM glycine, 3.0 μM vitamin B1, pH 5.5) containing 1 mM L-DOPA (L-3,4-dihydroxyphenylalanine), and were subsequently resuspended in 15 mL of minimal media with L-DOPA at a concentration of 10^7^ cells/mL. We incubated these cell suspensions at 37 °C in a shaking incubator for a duration of 5 days. Following the 5-day incubation period, we transferred 200 μL of cultures into a 96-well plate and photographed it at 600 dpi. To quantify melanization, we calculated the mean gray value for each well using ImageJ image processing software (ImageJ bundled with 64-bit Java 8, National Institutes of Health and the Laboratory for Optical and Computational Instrumentation, University of Wisconsin, Madison, WI, USA).

### 2.4. Assessment of Chitin Content in Cell Walls

To measure the chitin and chitosan content of *C. auris* wild-type and *tpk1*Δ *tpk2*Δ strains, we cultivated the cells in either YPD or melanin-inducing medium for a duration of 5 days, using a shaking incubator set to 37 °C. After cultivation, the cells were collected, washed, and resuspended in phosphate-buffered saline (PBS) with a pH value of 7.5. The cells were then subjected to staining using 25% CFW for 30 min in the dark, followed by two successive washes with PBS. Stained cells were visualized and documented using fluorescence microscopy (Olympus BX51: Tokyo, Japan). We quantified the fluorescence of at least 50 individual cells using ImageJ software (ImageJ bundled with 64-bit Java 8). 

### 2.5. Gene Deletion

We generated gene deletion mutants by utilizing the nourseothricin resistance marker (CaNAT), which was flanked by 0.3 to 0.7 kb 5′ and 3′ regions of each target gene, including B9J08_000072, B9J08_000073, B9J08_000517, and B9J08_002997. We constructed each gene disruption cassette, which contained a selection marker, through double-joint PCR. The first round of PCR amplified the flanking regions of a target gene using L1-L2 and R1-R2 primer pairs. The plasmid pV1025, containing the CaNAT gene, served as a template for amplifying the CaNAT selection marker by PCR, with primer pairs outlined in Appendix A. The first round of PCR products from the flanking regions and CaNAT marker were jointly purified and used as templates for the second round of double-joint PCR. In this round, we amplified 5′ and 3′ gene disruption cassettes, containing split CaNAT selection markers, by L1-split primer 2 and R2-split primer 1, respectively. 

We employed a lithium acetate/heat-shock protocol, with certain modifications, to transform *C. auris* with gene disruption cassettes. Cells were cultured overnight at 30 °C in 50 mL YPD broth under shaking conditions. An amount of 1.2 mL of cultured cells was centrifuged, washed with deionized water (dH_2_O) and lithium acetate buffer (100 mM lithium acetate, 10 mM Tris, 1 mM EDTA, pH 7.5), and then resuspended in 300 μL of the same buffer. The transformation was set up with 10 μL of denatured salmon sperm DNA (Sigma: St. Louis, MO, USA), 100 μL of competent cells, 500 μL of 50% PEG4000 (Sigma), and 50 μL of the amplified gene deletion cassette. The transformation mixture was subjected to a 6 h incubation at 30 °C with periodic vortexing, followed by a 20 min heat shock at 42 °C and 1 min cooling on ice. Subsequently, the cells were collected, resuspended in 1 mL of YPD medium, and incubated at 30 °C for 1 h with shaking. Post-incubation, the cells were washed twice with fresh YPD medium before being spread onto selective YPD agar plates containing 600 µg/mL nourseothricin. After a 2-day incubation at 37 °C, we verified the expected genotype of each positive nourseothricin-resistant transformant by diagnostic PCR and Southern blot (Appendix A). 

### 2.6. Assessing Survival Rate under Oxidative Stress

Yeast cells were cultured with or without L-DOPA at 37 °C for a duration of five days, then agitated and incubated in a MOPS (3-(N-Morpholino)propanesulfonic acid, 4-Morpholinepropanesulfonic acid)-buffered RPMI (Roswell Park Memorial Institute) medium. These cells were exposed to either 20 mM hydrogen peroxide (H_2_O_2_) or 0.08 mM menadione, both at 37 °C for a period of 3 h. Subsequently, the cell suspensions were diluted and spread on YPD agar plates to quantify the colony-forming units (CFU). The survival rate was determined by calculating the ratio of the yeast cell count under oxidative stress to the yeast cell count in non-stressed conditions.

## 3. Results

### 3.1. PKA Catalytic Subunits Are Involved in the Melanization of C. auris

Previous studies have reported the involvement of the cAMP/PKA signaling pathway in the synthesis of melanin in *Cryptococcus neoformans* [13,14]. Specifically, the disruption of adenylate cyclase *CAC1* and protein kinase *PKA1* impairs melanin production [13]. In our research, we investigated the relationship between the Ras/cAMP/PKA signaling pathway and melanization in *C. auris*. We evaluated melanin expression levels across various strains, including both wild-type and mutants. Interestingly, aside from the *tpk1*Δ *tpk2*Δ mutant strain, we observed no significant variation in melanin phenotype between the wild-type and other mutant strains (Figure 1A). In the culture medium, all mutant strains, except for *tpk1*Δ *tpk2*Δ, displayed a pronounced deep gray coloration. By contrast, the *tpk1*Δ *tpk2*Δ strain exhibited a lighter shade of gray (Figure 1A). Quantifying the gray intensity, mutant strains, excluding *tpk1*Δ *tpk2*Δ, ranged between 40 and 70, while the *tpk1*Δ *tpk2*Δ strain displayed a higher average gray intensity of around 120 (Figure 1B). When the culture medium of the wild-type strain was centrifuged, the resulting pellet was black. However, the pellet from the *tpk1*Δ *tpk2*Δ strain was dark gray (Figure 1C). Microscopic analysis demonstrated that under melanin-inducing conditions, the wild-type strain formed distinct, aggregated clumps robustly. By contrast, clump formation was compromised in the *tpk1*Δ *tpk2*Δ strain (Figure 1C). This suggests that the *tpk1*Δ *tpk2*Δ strain either failed to produce melanin granules effectively or had an issue with properly binding the produced melanin granules to the cell wall. 

To evaluate the attachment of melanin granules to the cell wall, we measured the supernatant absorbance from the culture medium of both the wild-type and mutant strains at 492 nm (Figure 1D). No statistically significant differences were found between the wild-type strain and other mutants. However, the *tpk1*Δ *tpk2*Δ strain exhibited a notable 84% increase in supernatant absorbance compared to the wild-type strain (Figure 1D). Taken together, our findings suggest that while the absence of Tpk1 and Tpk2 does not affect the synthesis of melanin granules in *C. auris*, it does impair their appropriate attachment to the cell wall. Furthermore, we provide evidence that the PKA catalytic subunits participate in the regulation of melanization in a manner independent of Cyr1.

### 3.2. Tpk1 and Tpk2 Regulate Chitin Synthesis under Melanin-Inducing Conditions

To further investigate the potential role of the cell wall in *C. auris* melanization, we examined the expression of genes implicated in chitin and chitosan synthesis, namely: *CDA2* (B9J08_004841), *CHS1* (B9J08_000158), *CHS2* (B9J08_005077), *CHS3* (B9J08_004150), *CHS4* (B9J08_004972), *CHS5* (B9J08_000868), *CHS6* (B9J08_004541), *CHS7* (B9J08_001816), and *CHS8* (B9J08_002856). We analyzed the expression levels of these genes at the 0, 6, and 24 h marks under melanin-inducing conditions in both the wild-type and *tpk1*Δ *tpk2*Δ strains. Under basal conditions, there was a negligible difference in gene expression levels between the wild-type and *tpk1*Δ *tpk2*Δ strains (Figure 2A). However, upon switching to melanin-inducing conditions, the expression patterns of cell-wall-related genes in the wild-type strain differed significantly from those in the *tpk1*Δ *tpk2*Δ strains. The wild-type strain showed a significant upregulation of chitin-synthesis-related genes, ranging from a 2.7- to an 11.7-fold increase compared to the basal condition, at the 24 h time point (Figure 2A). Conversely, the *tpk1*Δ *tpk2*Δ strain failed to properly induce the expression of chitin-related genes, except for *CHS5*, under melanin-inducing conditions, with a significant difference observed at the 24 h time point compared to the wild-type strain (Figure 2A). 

To evaluate the correlation between gene expression levels and chitin synthesis, we assessed the cell wall chitin content using calcofluor white (CFW) staining. Notably, the wild-type strain showed an increased fluorescence intensity upon CFW staining when cultured in melanin-inducing media, implying an elevated chitin content in the cell wall (Figure 2B,C). By contrast, the *tpk1*Δ *tpk2*Δ strain did not show a significant increase in fluorescence intensity compared to the wild-type strain when cultured in melanin-inducing media, suggesting a lack of significant increase in chitin content (Figure 2B,C). In summary, our findings indicate that while the inactivation of the Ras/cAMP/PKA signaling pathway in *C. auris* does not impair its capacity to produce melanin granules, the PKA catalytic subunits, Tpk1, and Tpk2, jointly control the expression of multiple chitin-synthesis-related genes under melanin-inducing conditions. This coordinated regulation of chitin synthesis is essential for the proper association of melanin granules with the cell wall.

### 3.3. Tpk1 Acts as the Principal Regulator of Melanization in C. auris

To discern which PKA catalytic subunit primarily influences melanization, we conducted melanin production experiments using Tpk1 and Tpk2 single knockout strains. The *tpk2*Δ strain exhibited a culture medium color akin to the wild-type strain, which was a dark brown (Figure 3A). By contrast, both the *tpk1*Δ and *tpk1*Δ *tpk2*Δ strains produced a lighter grayish color in the culture medium (Figure 3A). Upon measuring the gray value, we found that *tpk2*Δ exhibited a value of approximately 70, mirroring that of the wild-type strain, whereas *tpk1*Δ had a higher gray value of around 130 (Figure 3B). Examining the cell pellet color revealed that *tpk2*Δ had a black color, similar to the wild-type strain, while both *tpk1*Δ and *tpk1*Δ *tpk2*Δ exhibited a grayish color (Figure 3C). This suggests that *tpk1*Δ, much like *tpk1*Δ *tpk2*Δ, is incapable of properly binding melanin granules to the cell wall. Absorbance measurement of the supernatant indicated that *tpk1*Δ showed roughly a 50% increase in absorbance compared to the wild-type strain (Figure 3D).

Additionally, we measured the expression levels of chitin-synthesis-related genes under basal and melanin-inducing conditions (Figure 3E). The results revealed that most of the chitin-synthesis-related genes were not properly expressed in *tpk1*Δ. Overall, our findings suggest that Tpk1 plays a crucial role in melanization among the PKA catalytic subunits, while Tpk2 has a more minor role. The more pronounced melanization defect observed in *tpk1*Δ *tpk2*Δ compared to *tpk1*Δ implies a collaborative effort between Tpk1 and Tpk2 in regulating chitin synthesis in *C. auris*, which in turn impacts melanization.

### 3.4. Tpk1-Dependent Melanin Deposition Protects C. auris against Oxidative Stress

The role of melanin pigment as a potent antioxidant is crucial in protecting cells from oxidative stress. Our prior research demonstrated that *tpk1*Δ and *tpk1*Δ *tpk2*Δ mutants exhibit significantly increased susceptibility to oxidative stress when compared with the wild-type strain [15]. To explore the effect of melanization on *C. auris* resistance to oxidative stress, we examined the survival rates of both melanized and non-melanized versions of wild-type and *TPK1*/*2* mutant cells under oxidative stress conditions. 

In the wild-type strain, non-melanized cells showed an estimated survival rate of 70% after 3 h of exposure to 20 mM H_2_O_2_ and approximately 54% following a 3 h treatment with 0.08 mM menadione, a superoxide generator. By contrast, melanized cells displayed significantly enhanced survival rates of roughly 94% and 74%, respectively (Figure 4A,B). Comparable trends were observed in the *tpk2*Δ mutant. However, distinct patterns were observed in the *tpk1*Δ and *tpk1*Δ *tpk2*Δ mutants. With *tpk1*Δ, non-melanized cells exhibited survival rates of around 31% and 28% upon H_2_O_2_ and menadione treatment. Notably, melanized cells showed survival rates of approximately 42% and 38%, respectively, which did not significantly differ from non-melanized cells (Figure 4). Furthermore, the difference in survival rates between melanized and non-melanized cells in the *tpk1*Δ *tpk2*Δ mutants was only about 10%, implying that improper melanization in these mutants led to a lower variance in survival rates under oxidative stress conditions. These findings emphasize the crucial role of melanin granules as potent antioxidants, enabling *C. auris* to withstand oxidative stress effectively.

### 3.5. Multicopper Oxidases and Ferroxidases Are Dispensable for C. auris Melanization

Previous studies reported the presence of hypothetical multicopper oxidase and ferroxidase genes within the *C. auris* genome, yet their direct impact on *C. auris* melanization remained unexplored [12]. To investigate this, we constructed knockout strains for these genes and conducted melanization assays. Initially, we conducted a domain search for each gene to identify the associated domains or protein families. Consequently, we discovered that genes B9J08_000072 and B9J08_000073 contained a peroxisomal membrane protein domain, while genes B9J08_000517 and B9J08_002997 included a multicopper oxidase domain (Figure 5A). Subsequently, we analyzed the expression levels of these genes in both wild-type and *tpk1*Δ *tpk2*Δ strains under melanin-inducing conditions. Upon melanization induction in the wild-type strain, B9J08_000072, B9J08_000073, B9J08_000517, and B9J08_002997 showed approximately 4-fold, 8-fold, 5-fold, and 5-fold increases, respectively, compared to the basal condition (Figure 5B). In the *tpk1*Δ *tpk2*Δ strain, these genes also exhibited similar increases at around 3-fold, 5-fold, 3-fold, and 5-fold, respectively (Figure 5B). 

We next constructed knockout strains for the four target genes and conducted melanization assays (Appendix A). However, no melanization defects were detected in any of the deletion mutants, with melanization occurring to an extent similar to the wild-type strain (Figure 5C). Additionally, the supernatant absorbance measurements revealed values akin to those of the wild-type strain (Figure 5D). Overall, these results indicated a limited role for ferroxidase and multicopper oxidase in *C. auris* melanization, implying the involvement of other genes in melanin granule formation.

## 4. Discussion

In this study, we sought to explore the relationship between the Ras/cAMP/PKA signaling pathway and melanization in *C. auris*. Our investigations uncovered a pivotal role for the PKA catalytic subunit in the process of melanization in *C. auris*. Significantly, Tpk1 has a major impact on melanization in *C. auris*, while Tpk2 plays a minor role. We further established the role of Tpk1 in the melanization of *C. auris* through its regulatory control over the expression of several chitin-synthesis-related genes under melanin-inducing conditions.

Melanin is broadly recognized as a potent virulence factor in pathogenic fungi, serving to fortify the cell wall, perform antioxidant and UV-protective functions, and assist in evading host immune cells’ responses [16,17]. Although the functions, structure, and synthesis of melanin are yet to be fully understood, its various roles have been recognized, including its ability to protect fungal cells from external stress in pathogenic fungi such as *C. neoformans*, *Exophilia dermatitidis*, *Sporothrix schenckii*, *C. albicans*, and *Aspergillus fumigatus* [8]. Moreover, it has been shown that *C. auris* synthesizes black pigment melanin when cultured for five days in a medium containing l-3,4-dihydroxyphenylalanine (L-DOPA) as a substrate [12]. This underscores the necessity of investigating melanin as a virulence factor in *C. auris*. However, there remains a paucity of detailed research on the genes associated with melanin synthesis in *Candida* species. This gap in our understanding has made it challenging to investigate melanin synthesis in *C. auris*. 

In the extensively researched pathogenic fungus *C. neoformans*, genes involved in the Ras/cAMP/PKA signaling pathway have been found to play a role in melanin synthesis [13,14]. Specifically, it has been demonstrated that the deletion of adenylyl cyclase Cac1, along with the PKA catalytic subunits Pka1 and Pka2, results in a melanin-defective phenotype [13]. Consequently, drawing upon the findings in *C. neoformans*, we hypothesized that genes in the Ras/cAMP/PKA signaling pathway, particularly Cyr1 or Tpk1/Tpk2, could have a role in *C. auris* melanization. This led to the subsequent confirmation of a melanization defect in the *tpk1*Δ *tpk2*Δ mutant. However, in contrast with *C. neoformans*, it seems that the PKA catalytic subunit in *C. auris* plays a role in melanization that is independent of adenylate cyclase. This deduction is based on the observed melanization defects exclusively in the *tpk1*Δ *tpk2*Δ mutant, and not in the *cyr1*Δ mutant. In our previous study, we elucidated the diverse Cyr1-independent functions of PKA. Specifically, we observed significant differences between PKA deletion mutants and Cyr1 deletion mutants in ploidy switch, synthesis of cell wall components, biofilm formation, and virulence [15]. These findings underscore the pivotal role of PKA in these biological processes.

Furthermore, our discovery that the PKA catalytic subunit contributes to the synthesis of chitin, a major component of the cell wall, under melanin-inducing conditions, underscores its significant role in *C. auris* melanization. Various pathogenic fungi rely on chitin as an essential anchor for melanin production. This long-chain polymer, composed of β(1,4)-linked N-acetylglucosamine subunits, interacts with several cell wall proteins and polysaccharides [18]. The interaction between chitin and melanin was first elucidated in *Aspergillus nidulans*, and subsequently confirmed in *E. dermatitidis* and *C. neoformans* [16]. Consequently, we proposed that the chitin–melanin granule binding is also crucial in *C. auris*. Our experimental data corroborated this hypothesis by showing impaired expression of chitin-synthesis-related genes in the *tpk1*Δ and *tpk1*Δ *tpk2*Δ strains, both of which displayed melanization defects. Therefore, similar to other pathogenic fungi, the binding of melanin granules to chitin appears to be a critical step in *C. auris* melanization.

Nonetheless, our understanding of the specific genes implicated in melanin formation in *C. auris* still requires further elucidation. In our study, despite generating knockout strains for four putative melanin-synthesis-related genes and conducting melanization experiments, we found that none of these four genes significantly influenced melanization in *C. auris*. In the case of *C. neoformans*, melanin synthesis predominantly depends on laccase, with *LAC1* gene deletion resulting in pronounced melanin defects [19]. *C. albicans* does exhibit minor laccase activity compared to *C. neoformans*; however, this activity is relatively limited, and it remains inconclusive whether this directly influences melanin synthesis in *C. albicans*. Moreover, the genes involved in melanin synthesis have not been identified in *C. albicans*. Consequently, additional RNA-seq experiments under melanin-inducing conditions in *C. auris* appear necessary to pinpoint the genes participating in melanin granule synthesis.

In conclusion, our study sheds light on the critical role of the PKA catalytic subunit, particularly Tpk1, in *C. auris* melanization. This is achieved by regulating the expression of chitin-synthesis-related genes under melanin-inducing conditions. However, future research is mandated to discern the specific genes engaged in melanin formation in *C. auris*, given that our study did not find a direct association with the four putative melanin-synthesis-related genes examined.

## Figures and Tables

**Figure 1 antioxidants-12-01702-f001:**
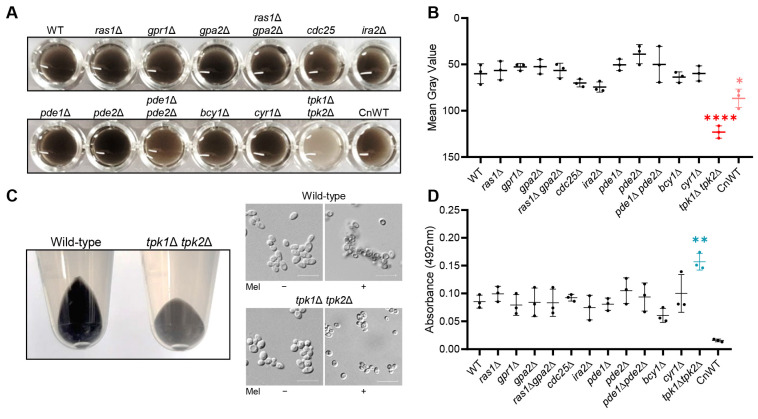
The role of the Ras/cAMP/PKA pathway in *C. auris* melanization. (**A**) Comparative analysis of melanization in *C. auris* wild-type and Ras/cAMP/PKA mutant strains. *Cryptococcus neoformans* H99 strain was used as a melanizing control species (CnWT). Wild-type and mutant strains were cultured in a melanin-inducing medium for 5 days to induce melanization, followed by imaging of the cell suspension. This experiment was independently repeated three times, and a representative image from one trial is shown. (**B**) The gray values of three biologically independent images were measured using Image J. (**C**) After a 5-day incubation of wild-type and *tpk1*Δ *tpk2*Δ strains on a melanin-inducing medium, we collected the cells via centrifugation. Images of the resulting cell pellet were captured, and cell morphology was observed using microscopy with a DIC filter. Scale bar = 10 μm. (**D**) After 5 days of melanization, the absorbance of the supernatant was measured. (**B**,**D**) Error bars represent standard deviation. Statistical analysis was performed using one-way ANOVA with Bonferroni’s multiple comparison test (* *p* < 0.05; ** *p* < 0.01; **** *p* < 0.0001).

**Figure 2 antioxidants-12-01702-f002:**
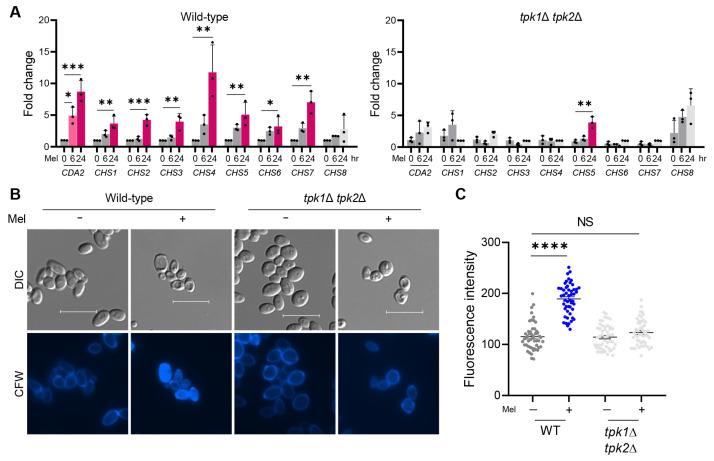
Chitin synthesis in wild-type and *tpk1*Δ *tpk2*Δ strains under melanin-inducing conditions. (**A**) Quantitative reverse transcription PCR (qRT-PCR) analysis of chitin- and chitosan-related genes in wild-type and *tpk1*Δ *tpk2*Δ strains at each time point. Three independent biological experiments with three technical replicates were performed. (“Mel” refers to the Melanin-inducing medium.) (**B**) Chitin staining. The wild-type and *tpk1*Δ *tpk2*Δ strains were cultured on YPD or melanin-inducing medium for 5 days and then stained with FITC-conjugated calcofluor white (CFW). Scale bar = 10 μm. (**C**) Quantitative fluorescence measurements of at least 50 individual cells of each strain, measured using ImageJ software (ImageJ bundled with 64-bit Java 8, National Institutes of Health and the Laboratory for Optical and Computational Instrumentation, University of Wisconsin, WI, USA), are shown. Three independent biological experiments with three technical replicates were performed. (**A**,**C**) Mean values are shown with error bars indicating standard deviation. Statistical analysis was performed using one-way ANOVA with Bonferroni’s multiple comparison tests (* *p* < 0.05; ** *p* < 0.01; *** *p* < 0.001; **** *p* < 0.0001; NS, not significant).

**Figure 3 antioxidants-12-01702-f003:**
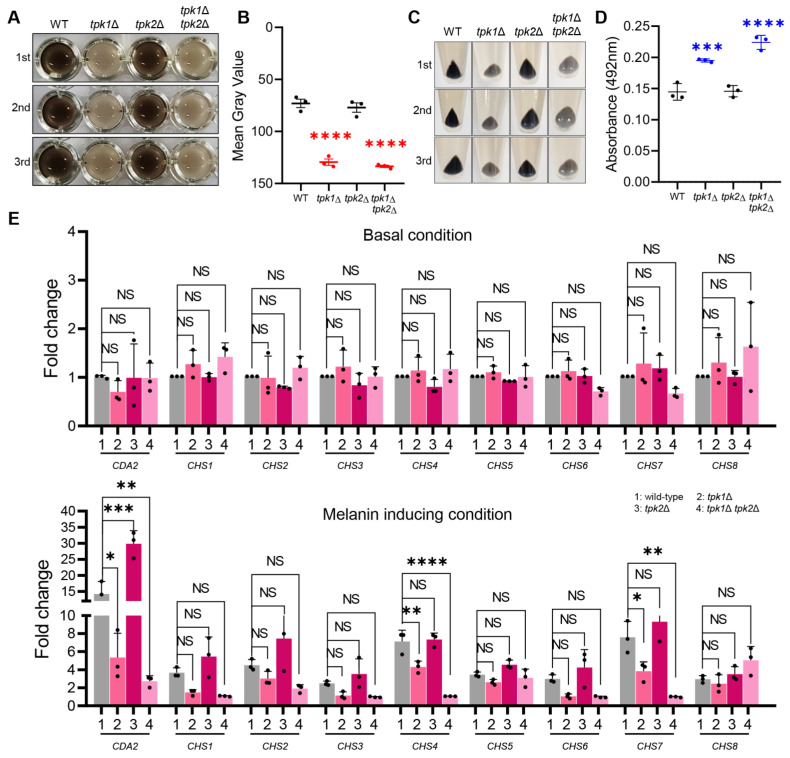
The primary role of Tpk1 in *C. auris* melanization. (**A**) Comparison of melanization between *C. auris* wild-type strain and PKA catalytic subunit mutant strains. (**B**) The mean gray values of three biologically independent images were measured using Image J. (**C**) Image of the pellet from melanized cells. (**D**) After 5 days of melanization, the absorbance of the supernatant was measured. (**E**) Quantitative reverse transcription PCR (qRT-PCR) analysis of chitin- and chitosan-related genes in wild-type, *tpk1*Δ, *tpk2*Δ, and *tpk1*Δ *tpk2*Δ strains at each time point (0, 24 h). Three independent biological experiments with three technical replicates were performed. (**B**,**D**,**E**) Mean values are shown with error bars indicating standard deviation. Statistical analysis was performed using one-way ANOVA with Bonferroni’s multiple comparison tests (* *p* < 0.05; ** *p* < 0.01; *** *p* < 0.001; **** *p* < 0.0001; NS, not significant).

**Figure 4 antioxidants-12-01702-f004:**
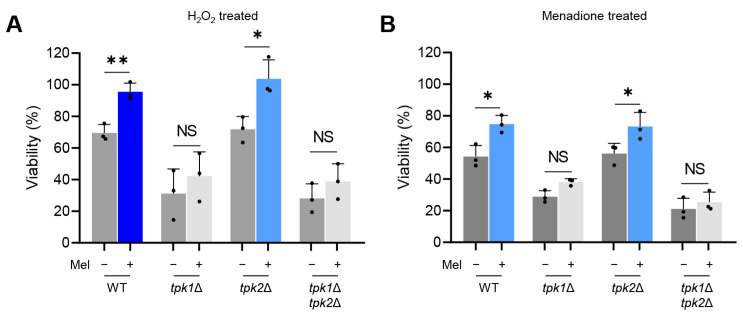
Tpk1-dependent melanin deposition is crucial for *C. auris* to resist antioxidants. Melanized and control yeast cells were incubated for 3 h at 37 °C in the presence or absence of (**A**) 20 mM hydrogen peroxide or (**B**) 0.08 mM menadione. The viability of *C. auris* was assessed by counting colony-forming units (CFUs), and the results for H_2_O_2_- or menadione-treated cells are expressed as a percentage relative to the cells without H_2_O_2_ or menadione. (**A**,**B**) Error bars indicate standard deviation. Statistical analysis was performed using Student’s *t*-test (* *p* < 0.05; ** *p* < 0.01; NS, not significant).

**Figure 5 antioxidants-12-01702-f005:**
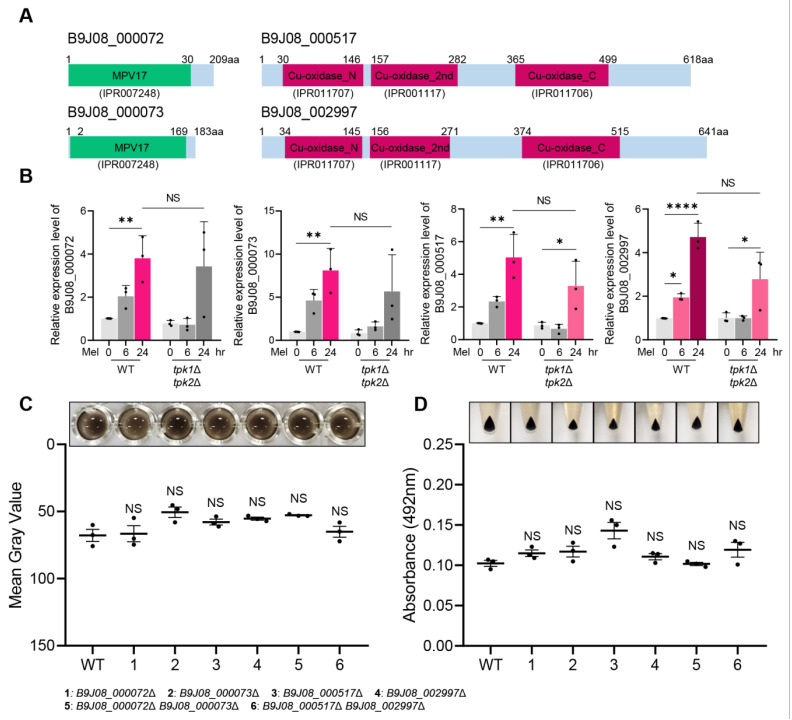
Roles of multicopper oxidases and ferroxidases in *C. auris* melanization. (**A**) Protein domain analysis of putative melanin-synthesis-related genes in *C. auris*. This domain analysis image was created based on the domain information available on InterPro (https://www.ebi.ac.uk/interpro/, accessed on 11 June 2023). (**B**) Quantitative reverse transcription PCR (qRT-PCR) analysis of multicopper oxidase and ferroxidase genes in wild-type and *tpk1*Δ *tpk2*Δ strains at each time point (0, 6, and 24 h). Three independent biological experiments with three technical replicates were performed. (**C**) The mean gray values of three biologically independent images were measured using Image J. (**D**) After 5 days of melanization, the absorbance of the supernatant was measured. (**B**–**D**) Error bars indicate standard deviation. Statistical analysis was performed using one-way ANOVA with Bonferroni’s multiple comparison test (* *p* < 0.05; ** *p* < 0.01; **** *p* < 0.0001; NS, not significant).

## Data Availability

The data are contained within this article.

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
