# Peer review of "Protein Kinase A Controls the Melanization of Candida auris through the Alteration of Cell Wall Components"

_antioxidants, 2023, doi:10.3390/antiox12091702_

Round 1

Reviewer 1 Report

The authors of the MS entitled, “Protein Kinase A Controls the Melanization of Candda auris through Alteration of Cell Wall Component” provide information on the role of melanin in the protection of C. auris cell integrity.

 Overall is a well-written paper and provides new information on the  way melanin protects this species from oxidation by decreasing the related chitin synthesis gene expression, e.g., Tpk1.

 However, most figure legends are too long, as only the important issues should be pointed out to the reader. It also could be easier to see differences found under the text, Results and Discussion, instead of as figure legends.

The paper is well written.

Reviewer 2 Report

This manuscript described by Kim and Bahn showed the role of Protein kinase A termed Tpk1 and Tpk2 in melanization of Candida auris. They showed Tpk1 plays a major role in melanization, chitin synthesis and tolerane to oxidative stress. The also claimed Tpk1 and Tpk2 are separately regulated from the RAS-cAMP pathway.

1) Many gene names needs to be italicized. Check through out the text.

2) Figure 1, it is not stated what is H99. It needs explanation.

3) I assume cda2 is one of chitin deacetylase to synthesize chitosan. This explanation is necessary. They analyzed the expression of the cda2 gene. Is there cda1 in Candida auris?

4)In figure 2 they showed the CHS5 gene is induced in the tpk1 tpk2 double deletion strain. In figure 3, they do not state about this point. Is this a repeatable observation?

5) I do not believe chitin is critical for melanization unless the authors derived the mutant defective in chitin synthase. Calcofluor does not only stain chitin, it also stains other polysaccharides.

6) L 226  ‘the inactivation of the Ras/cAMP/PKA signaling pathway in C. auris does not impair its capacity to produce melanin granules,--- ‘

----This expression is confusing because the authors claimed PKA is separately regulated from cAMP pathway. Another point; Is there evidence Ras is controlling AC in C. auris? Is this inferred from the knowledge on C. albicans?

7) L 347 Overall, these results a limited role for ferroxidase and multicopper oxidase in C. auris melanization, implying

---This statement needs a paraphrase.
